# Purification and Properties of Polyphenol Oxidase of Dried *Volvariella bombycina*

**DOI:** 10.3390/biology12010053

**Published:** 2022-12-28

**Authors:** Assemgul Sarsenova, Dudu Demir, Kardelen Çağlayan, Sardarbek Abiyev, Talshen Darbayeva, Cafer Eken

**Affiliations:** 1Department of General Biology and Genomics, Faculty of Natural Sciences, L.N. Gumilyov Eurasian National University, Satpayev str. 2, Astana 010000, Kazakhstan; 2Department of Biology and Ecology, Faculty of Natural Geography, Makhambet Utemisov West Kazakhstan University, N. Nazarbayev ave.162, Uralsk 090000, Kazakhstan; 3Department of Agricultural Biotechnology, Faculty of Agriculture, Isparta University of Applied Sciences, 32260 Isparta, Türkiye; 4Department of Agricultural Biotechnology, Faculty of Agriculture, Aydın Adnan Menderes University, 09070 Aydın, Türkiye

**Keywords:** *Volvariella bombycina*, polyphenol oxidase, affinity chromatography, purification, characterization

## Abstract

**Simple Summary:**

Mushrooms are non-green, edible fungi that are high in non-starchy carbohydrates, fiber, protein, minerals, and vitamins. Mushrooms have over a hundred different medicinal properties. *Volvariella bombycina*, an important wild, edible, and medicinal mushroom, has commercial cultivation potential. Polyphenol oxidases (PPOs), copper-containing metalloprotein enzymes that are widely distributed in microorganisms, plants, and animals, are responsible for melanization in animals and browning in plants. Mushrooms are commonly used as a potent PPO source because they can be obtained in relatively large amounts and are cheap. PPO was purified and characterized from dried *V*. *bombycina*, and it will also encourage researchers to perform further studies. PPO was purified and characterized for the first time from a dried wild edible and medicinal mushroom (*V*. *bombycina*).

**Abstract:**

Polyphenol oxidase (PPO) was purified and characterized from a dried wild edible and medicinal mushroom (*V*. *bombycina*). Using Sepharose 4B-L-tyrosine-*p*-aminobenzoic acid affinity chromatography, PPO was purified from the dried *V*. *bombycina.* The purification was completed with a 33.85-fold purification. On sodium dodecyl sulfate-polyacrylamide gel electrophoresis (SDS-PAGE), the purified enzyme migrated as a single band. The molecular weight of the purified enzyme was estimated by SDS-PAGE to be about 25 kDa. Catechol, 4-methyl catechol, and pyrogallol were used as substrates to determine the enzyme activity and its kinetic parameters (*Km* and *Vmax*). At the optimum pH and temperature, dried *V*. *bombycina* PPO’s *Km* and *Vmax* values for catechol, 4-methyl catechol, and pyrogallol were found to be 1.67 mM–833.33 U/mL, 3.17 mM–158.73 U/mL, and 2.67 mM–3333.33 U/mL, respectively. Also investigated were the effects of pH and temperature on the enzymatic properties of PPO in dried *V*. *bombycina*. The optimum pH and temperature values for dried *V*. *bombycina* PPO obtained by using catechol, 4-methyl catechol, and pyrogallol as substrates were 6.5, 15 °C; 9.0, 20 °C; and 8.0, 15°C, respectively. This is the first study on the purification and characterization of PPO from dried *V*. *bombycina*.

## 1. Introduction

Mushrooms are characterized as the macroscopic fruiting bodies of fungi and are a nutritious food source as well as a source of physiologically beneficial and nontoxic medicines [1]. The majority of edible mushrooms are Basidiomycota, except truffles and morels, which are Ascomycota [2,3]. Mushrooms contain a good supply of non-starchy carbohydrates, dietary fiber, protein, minerals, and vitamins [4]. They are precious products, richer in some elements than most veggies. The most common method of mushroom processing is drying, and dried mushrooms are mostly used for home consumption or food service as well as as a food industry ingredient: as a spice mix or in dehydrated soups. In addition to being a food, macrofungi have antitumor or anticancer properties, as well as antioxidant and antimicrobial properties [5]. Many of them have been used in folk medicine for thousands of years.

Among such, one important wild edible and medicinal mushroom, *V. bombycina* (Schaeff.) Singer is commonly known as the silky sheath, silky agaric, silky rosegill, silver-silk straw mushroom, or tree mushroom, and has the potential for commercial cultivation [6,7]. It belongs to the family Pluteaceae of Basidiomycota and has been reported from Australia, Asia, the Caribbean, North America, and Europe [7,8,9,10,11]. Moreover, it has been reported that several compounds isolated and identified from *V*. *bombycina* exhibit biological activities, including antitumor, antioxidant, and hypercholesterolemic effects [10,12,13].

Enzymes are present in every living organism, including mushrooms, and are used for hydrolysis, oxidation, reduction, and metabolism [14]. It is essential for mushroom development, nutritive value, and flavor. A mushroom species’ ability to produce hydrolytic enzymes has a direct impact on its ability to produce mushrooms. The fruiting body of the mushroom is typically white at first, but during food processing, the mushroom may occasionally develop brown to black color changes as a result of the activity of enzymes such as polyphenol oxidase (PPO). PPO is a copper-containing enzyme belonging to the family of oxidoreductases, classified into EC1.10.3.1 (catechol oxidase, diphenol oxidase, or *o*-diphenol oxygen oxidoreductase) and EC1.14.18.1 (cresolase, monophenol monooxygenase, or tyrosinase). Polyphenol oxidases are a widespread group of enzymes found in animals, bacteria, fungi, and plants, especially fruits and vegetables [15]. The main roles of plant PPOs are in enzymatic browning processes and biotic and abiotic stress defense for organisms [16]. In addition, PPO is commonly utilized in several fields, including healthcare and medicine. It is now the focus of attempts to produce new medications to treat Alzheimer’s and Parkinson’s diseases [17]. In a recent study, a new anticancer drug PPO was created by extracting from edible mushroom. It showed good cancer resistance to ovarian, lung, breast, and prostate cancers [17]. Melanogenesis is the complicated process by which the pigment melanin is produced in melanosomes by melanocytes. The primary enzyme in melanin manufacture is known as PPO (tyrosinase) and it plays an important role in melanogenesis [18]. Tyrosine-related melanogenesis plays a significant role for pigmentation of eye, hair, and skin in mammals, as pigmentation is a pivotal part of skin protection by UV radiation [19]. In mushrooms (Basidiomycota), PPOs have been well researched because of the undesirable postharvest enzymatic browning they cause, which downgrades the value of these products. Amongst mushrooms, *Agaricus bisporus* is the most studied species worldwide for PPO [5,18,20,21,22,23,24,25,26]. PPO has also been isolated and characterized from various mushroom species such as *Amanita muscaria* [27], *Armillaria mellea* [28], *Armillaria ostoyae* [29], *Boletus erythropus* [30], *Hypholoma fasciculare* [28], *Lactarius pergamenus* [31], *Lactarius piperatus* [32], *Lactarius salmonicolor* [33], *Lentinula boryana* [34], *Lentinus edodes* [35,36], *Lepiota procera* [37], *Lepista nuda* [28], *Macrolepiota gracilenta* [38], *Pleurotus djamor* [39], *Pleurotus ostreatus* [40], *Russula delica* [41], and *Volvariella volvacea* [42]. However, there are no reports on the PPO enzymes from *V*. *bombycina*.

The extraction, purification (ammonium sulfate precipitation, dialysis, and affinity column chromatography), and characterization of PPO from dried *V*. *bombycina* were the aims of this study. After the enzyme had been purified using Sepharose 4B-L-tyrosine-*p*-aminobenzoic acid affinity chromatography, its biochemical characteristics, including its kinetic properties (*Km* and *Vmax*), optimum temperature and pH range, and substrate specificity, was determined. SDS-PAGE gel electrophoresis was carried out for molecular weight determination of the dried *V*. *bombycina* PPO enzyme.

## 2. Materials and Methods

### 2.1. Collection of Samples

Samples of *V. bombycina* were collected from the trunk of European white elm trees (*Ulmus laevis*) found at the Ural River Valley within West Kazakhstan (N50° 26.043′ E51° 08.494′). They were put in privately prepared boxes and brought to the laboratory. Macromorphological data were obtained from fresh samples, and micromorphological features were examined under a light microscope and identified [43,44]. *V*. *bombycina* as fruiting bodies (pileus + stipe) were dried under natural light at room temperature for several days. The herbarium was prepared and deposited at the Herbarium Laboratory, Department of General Biology and Genomics, L.N. Gumilyov Eurasian National University, Astana, Kazakhstan, and Herbarium V. V. Ivanov, Department of Biology and Ecology, Faculty of Natural Geography, M. Utemisov West Kazakhstan University, Uralsk, Kazakhstan.

### 2.2. Chemicals

Cyanogen bromide solution (CNBr), sepharose 4B, *p*-aminobenzoic acid, L-tyrosine, L-ascorbic acid, sodium dodecyl sulfate (SDS), dialysis sack (dialysis tubing cellulose membrane avg. flat width 25 mm (1.0 in.)), sodium bicarbonate (NaHCO_3_), ammonium persulfate, pyrogallol, 4-methyl catechol, 30% solution of acrylamide/bis-acrylamide, N,N,N′,N′-tetramethyl ethylenediamine (TEMED), and glycine were purchased from Sigma-Aldrich Co. Ltd. (Saint Louis, MO, USA). Tris (hydroxymethyl)aminomethane (TRIS), polyethylene glycol 4000 (PEG), di-potassium hydrogen phosphate (K_2_HPO_4_), di-sodium hydrogen phosphate (Na_2_HPO_4_), sodium hydroxide (NaOH), hydrochloric acid 37% (HCl), sodium chloride (NaCl), catechol, β-mercaptoethanol, sodium nitrite (NaNO_2_), ammonium sulfate ((NH_4_)_2_SO_4_), bovine serum albumin (BSA), ortho-phosphoric acid 85% (H_3_PO_4_), glycerol, acetic acid (CH_3_CO_2_H), Triton X-100, and Coomassie Brilliant blue G 250 were purchased from Merck (Darmstadt, Germany). Coomassie Brilliant Blue R-250 was purchased from Amresco Inc. (Solon, OH, USA). A protein molecular weight marker was purchased from Thermo Scientific (Vilnius, Lithuania). Bromophenol blue was purchased from Fisher Scientific (UK). Ethanol was purchased from İnterlab Laboratory Products Industry and Trade Limited Company (Istanbul, Turkey). Methanol was purchased from J.T. Baker (The Netherlands).

According to Arslan et al. [45], the Sepharose 4B-L-tyrosine-*p*-aminobenzoic acid affinity gel used in this study for the purification of PPO was synthesized in the Enzyme and Microbial Biotechnology laboratory of Isparta University of Applied Sciences (Isparta, Turkey).

### 2.3. Preparation of Crude PPO Extract

The PPO enzyme extraction was performed according to Saki et al. [37] with some modifications. Five grams of dried *V*. *bombycina* samples were crushed by liquid nitrogen and homogenized for 1–2 min in a Waring laboratory blender with 50 mL extraction buffer (100 mmol/L, pH 7) which contained 0.87 g K_2_HPO_4_, 0.5% (*w*/*v*) PEG, 1 mmol/L ascorbic acid, and 1% (*v*/*v*) Triton X-100. The buffer is freshly prepared. After extraction, the mixture was filtered with cheesecloth and then centrifugated at 15,000 rpm for 30 min at 4 °C. The supernatant was stored in a refrigerator and used as a crude enzyme source for further experiments.

### 2.4. Purification of PPO from Dried V. bombycina

#### 2.4.1. Precipitation with Ammonium Sulfate

Precipitation with ammonium sulfate was used as a purification step for the soluble, crude PPO. The crude extract was fractionated and performed according to Arslan et al. [45] with some modifications. By gradually adding solid ammonium sulfate and continually stirring with a magnetic stirrer until it was all dissolved, the crude enzyme was raised to 80% ammonium sulfate saturation. The precipitate was obtained after centrifuging the suspension at 15,000 rpm for 40 min at 4 °C. In phosphate buffer (5 mmol/L, pH 6.3), the precipitate was redissolved.

#### 2.4.2. Dialysis

The sample that was obtained from ammonium sulfate precipitation was dialyzed (phosphate buffer (5 mmol/L, pH 6.3) through the membrane overnight at 4 °C, changing the buffer. The solution of the dialyzed enzyme was used for subsequent experiments [45].

#### 2.4.3. Affinity-Column Chromatography of PPO

PPO enzymes were purified from dried *V*. *bombycina* using a Sepharose 4B-L-tyrosine-*p*-aminobenzoic acid affinity column. The enzyme solution was applied to the affinity column (2.5 × 10 cm), which was pre-equilibrated with 50 mM phosphate buffer solution (pH 5.0). The same buffer solution was used to wash the affinity gel (40 mL). PPO was eluted with a 0.05 mol/L phosphate buffer solution (pH 7.0) including 1 M NaCl. The PPO was collected using Eppendorf tubes.

#### 2.4.4. Determination of Protein Concentration

The protein concentration of each collected tube was determined according to the method described by Bradford [46], using bovine serum albumin as a standard. The amount of purified protein obtained by chromatography was determined by measuring absorbance at 595 nm.

#### 2.4.5. Determination of Dried *V. bombycina* PPO Activity

PPO activity was measured using catechol as a substrate, based on the rise in absorbance at 420 nm. Measures of 0.83 mL of 0.1 M phosphate buffer solution, 0.13 mL of 0.1 M substrate, and 0.04 mL of the enzyme solution were all present in the sample cuvette. Only 1.0 mL of substrate solution was present in the blank sample. The amount of enzyme that resulted in a change in absorbance of 0.001 mL^−1^ min^−1^ was defined as one unit of PPO activity [45]. PPO activity measurements were all assayed in triplicate.

### 2.5. Physicochemical Properties of PPO

#### 2.5.1. Determination of Dried *V. bombycina* PPO Substrate Specificity

A spectrophotometric method was used to assess the PPO activity of dried *V*. *bombycina* [43] utilizing three distinct substrates (catechol, 4-methyl catechol, and pyrogallol). At corresponding wavelengths of 420 nm (catechol and 4-methyl catechol) and 320 nm (pyrogallol), the PPO activity was determined by the standard assay procedure. The amount of enzyme that caused a change in absorbance of 0.001 mL^−1^ min^−1^ was used to define one unit of PPO activity [45].

#### 2.5.2. Effects of pH on Dried *V. bombycina* PPO Activity

The optimum pH for the dried *V*. *bombycina* PPO was performed according to Arslan et al. [45] with some modifications. The optimum pH for the PPO was determined in 0.1 mol/L sodium acetate buffer (pH 4.5–5.5), phosphate buffer (pH 6.0–8.0), and Tris-base (pH 8.5–9.0) at different temperatures in the range 15–45 °C with using three different substrates (0.1 mol/L catechol, 0.1 mol/L 4-methyl catechol and 0.1 mol/L pyrogallol). The assays were performed in triplicate.

#### 2.5.3. Effects of Temperature on Dried *V. bombycina* PPO Activity

The optimum temperature for the dried *V*. *bombycina* PPO was determined to range from 15–45 °C using the three different substrates (0.1 mol/L catechol, 0.1 mol/L 4-methyl catechol and 0.1 mol/L pyrogallol), and 10 different pH values ranging from 4.5–9 according to Arslan et al. [45] with some modifications. The assays were performed in triplicate.

#### 2.5.4. Electrophoresis

To determine the molecular weight of the purified enzyme, sodium dodecyl sulfate-polyacrylamide gel electrophoresis (SDS-PAGE) was conducted according to the Laemmli method [47]. In a Mini Protean Tetra Cell Electrophoresis Unit (Bio-Rad), with 10% running gel and 3% stacking gels, purified PPO was subjected to SDS–PAGE. Purified PPO (approximately 50 μg/mL) was loaded in each lane, and the slab gels (1 mm thickness) were run at 80 volts in the stacking gel and 150 volts in the running gel. After electrophoresis, protein bands were made visible using Coomassie Brilliant Blue R-250. The molecular weight of PPO was estimated with the protein molecular weight marker (116.0, 66.2, 45.0, 35.0, 25.0, and 18.4 kDa).

#### 2.5.5. Kinetic Properties of Dried *V. bombycina* PPO

The Michaelis–Menten constant (*Km*) and maximum velocity (*Vmax*) values for dried *V*. *bombycina* PPO were determined by measuring the enzyme activity at varying concentrations of substrates for catechol, 4-methyl catechol, and pyrogallol at optimum pH and temperature by using Lineweaver–Burk’s plots [48]. Measurements were performed in triplicate.

## 3. Results and Discussion

### 3.1. Purification Profile of PPO of Dried V. bombycina

There is no report describing PPO from dried *V*. *bombycina*, despite the fact that PPO has been purified and characterized from numerous mushroom species. In this study, polyphenol oxidase was purified from dried *V*. *bombycina* with affinity chromatography. By using the Sepharose-4B-L-tyrosine-*p*-aminobenzoic acid affinity column, the PPO enzyme was purified from dried *V*. *bombycina*. Affinity chromatography on immobilized (through 4-aminobenzoic acid) tyrosine-Sepharose is the more effective method [30]. In addition, chromatographic methods provide a high degree of purification [49]. Affinity chromatography is a practical method that is mostly used among chromatographic methods [50].

To investigate the properties of PPO from dried *V*. *bombycina*, we purified the enzyme by ammonium-sulfate precipitation and four chromatography steps, as summarized in Table 1. As shown in Table 1, the PPO enzyme purified 33.85 folds from dried *V*. *bombycina* using Sepharose 4B-L-tyrosine-*p*-aminobenzoic acid affinity chromatography. The content of total protein decreased 1370-fold in this process, and the total activity of the enzyme decreased by 40.3 times. In the previous study using the same affinity chromatography gel, PPO was purified 28.5-fold from *Boletus erythropus* [30], 65.3-fold from *Macrolepiota gracilenta* [38], 13.9-fold from *Lactarius piperatus* [32], and 26.6-fold from *Lactarius salmonicolor* [33]. In the studies using different chromatographic methods, PPO was purified 16.36-fold [18] and 32.05-fold [26] from *A*. *bisporus*, 51.7-fold [39] from *Pleurotus djamor* by utilizing a DEAE-cellulose column chromatography, 49-fold from *Pholiota nameko* by using Superdex 200 prep grade column [51], and 41.9-fold from *A*. *bisporus* by utilizing DEAE–Sepharose Fast Flow chromatography and Phenyl Sepharose CL-4B chromatography [5].

Experimental conditions have an impact on PPO activity, and because it is sensitive to extraction techniques, different preparation techniques will vary from each other [52]. Ahlawat et al. [42] reported that among the twelve *Volvariella volvacea* strains, the highest PPO activity was recorded with *V. volvacea* strain OE-210 using catechol as substrate. The levels of PPOs are well known to be dependent on the substrate, species, cultivar, maturity, and age [53]. Phenolic compounds are the primary substrates of PPO. In this study, the determination of the PPO activity used pyrogallol, 4-methyl catechol, and catechol as substrates. Although catechol is used as the most reacting substrate, other compounds such as 4-methyl catechol, pyrogallol, catechine, etc. are also widely studied [49].

SDS-PAGE electrophoresis was used to evaluate the PPO from dried *V*. *bombycina* that was eluted in the chromatographic separations. Figure 1 shows the protein staining result. SDS-PAGE of PPO successfully purified from dried *V. bombycina* using an affinity column revealed only a single protein band corresponding to a molecular weight of approximately 25 kDa.

Following is a report on the molecular weight of PPO from various mushroom species as determined by denaturing SDS-PAGE: *A*. *bisporus*, 95 kDa [18], 70 kDa [23], 43 kDa [5,26]; *Armillaria ostoyae*, 66 kDa [29]; *Boletus erythropus*, 40 kDa [30]; *Lactarius pergamenus*, 64 kDa [31]; *Lactarius piperatus*, 40 kDa [32]; *Lactarius salmonicolor* 65 kDa [33]; *Lepiota procera*, 35 kDa [37]; *Macrolepiota gracilenta*, 57.3 kDa [38]; *Pholiota nameko*, 42 kDa [51], and *Pleurotus djamor*, 90 kDa [39]. Our results indicate that the molecular weight of dried *V*. *bombycina* PPO is smaller than that of other mushroom PPOs.

### 3.2. Effects of pH and Temperature on Dried V. bombycina PPO Activity

Enzyme activity depends largely on environmental factors such as pH and temperature. The pH plays an important role in determining the statement of enzymatic activity. Temperature is another important factor that influences enzyme catalytic efficiency. The optimum temperature for enzymatic activity generally depends on the conditions of an experiment. Due to thermal denaturation, reactions’ rates usually decrease at high temperatures. The activity of dried *V*. *bombycina* PPO was measured at different pHs and temperatures using catechol, 4-methyl catechol, and pyrogallol as substrates. The pH profile of dried *V. bombycina* PPO activity was determined between 4.5 and 9.0 with 0.5 pH unit intervals. The effect of temperature on dried *V*. *bombycina* PPO activity was also investigated in the range of 15–45 °C. The enzymatic profile of the dried *V*. *bombycina* PPO at different pH values and temperatures is shown in Figure 2. The optimal pH and temperature values of the PPO of the dried *V*. *bombycina* were estimated by using catechol, 4-methyl catechol, and pyrogallol as substrates, and the results are shown in Table 2. The optimal pH and temperature values for the PPO enzyme of dried *V*. *bombycina* with catechol, 4-methyl catechol, and pyrogallol as substrates were 6.5, 15 °C; 9.0, 20 °C; and 8.0, 15 °C, respectively. The optimal pH and temperatures of dried *V*. *bombycina* PPO changed depending on the substrate used (Table 2). The highest PPO activity for dried *V*. *bombycina* was found at 26,692.50 U/mL at pH = 8 and the temperature at 15 °C using pyrogallol as substrate (Table 2 and Figure 2). In previous reports, it was reported that various plants have different optimum pH values. It had previously been reported that the optimum pH was 7 for *A*. *bisporus* [54] and *L. piperatus* [32] PPOs, 5 for *M*. *gracilenta* [38], and 6 for *L*. *pergamenus* [31] using catechol as the substrate. The literature has reported various optimum temperatures for PPO obtained from various sources. The optimum temperature of 20 °C had earlier been reported for *L. piperatus* PPOs [32], 35 °C for *A*. *bisporus* [26], and 10 °C for *L. pergamenus* [31] using catechol as the substrate. Moreover, it has been reported that the optimum temperature and pH for the *A*. *bisporus* PPO activity were 20 °C and a pH of 6.5–7.0 using catechol as the substrate [5].

For *A*. *mellea* and *H*. *fasciculare* PPOs, using 4-methyl catechol as the substrate, the optimum temperatures were previously found to be 30 °C and 20 °C, respectively [28]. Gouzi and Benmansour [55], using a pyrogallol substrate, determined two pH optimums levels of *A*. *bisporus* PPO activity, 5.3 and 7.0 at 25 °C. Dedeoğlu and Guler [33] reported that the optimum pH values of *Lactarius salmonicolor* PPO are 7.5, 7.5, and 6 for catechol, pyrogallol, and 4-methyl catechol, respectively. The optimum pH for PPO generally ranges from 4.0 to 8.0 but varies greatly depending on the plant source [56]. Because of this, it is obvious that the optimum pH and temperatures for PPO depend on the species and substrate.

### 3.3. Kinetic Parameters

The Michaelis–Menten constant *Km*, the, maximum reaction velocity *Vmax*, and the value of *Vmax/Km* of PPO in dried *V*. *bombycina* were calculated using Lineweaver–Burks plots under optimum pH and temperature conditions (Figure 3). The reported *Km* values for the PPO enzyme are usually in the range of 1–10 mM, indicating that, in general, it has a low affinity for its substrates [57]. A decrease in the *Vmax*/*Km* ratio and an increase in the *Km* value indicates that the residual enzyme’s affinity for its substrate has decreased. Catechol, 4-methyl catechol, and pyrogallol were utilized as substrates in this study, and their oxidation rates were determined. Table 3 shows the kinetic properties of the PPO enzyme. *Km* and *Vmax* of dried *V*. *bombycina* PPO using catechol, 4-methyl catechol, and pyrogallol as substrates were determined to be 1.67 mM, 833.33 U/mL, 3.17 mM, 158.73 U/mL, and 2.67 mM, 3333.33 U/mL, respectively. Among the three substrates, catechol was shown to have the lowest *Km* value toward dried *V*. *bombycina* PPO (*Km* of 1.67 mM). According to these results, catechol had a greater affinity than the other substrates (4-methyl catechol and pyrogallol). The highest *Vmax* values of PPO from dried *V*. *bombycina* were determined to be 3333.33 U/mL using pyrogallol as the substrate. The *Vmax*/*Km* ratio, which is greater in pyrogallol, is the determining criterion for the best substrate (Table 3). The kinetics of PPO in several mushroom species have been investigated in many studies utilizing catechol, 4-methyl catechol, and pyrogallol as substrates. The *Km* values of PPO from *A*. *bisporus* are: 0.67 mM [5], 0.71 mM [26], 2.1 mM [54] for catechol, and 1.4 mM [55], 2.16 mM [58] for pyrogallol. Öz et al. [32] found a *Km* for catechol of 1 mM using *Lactarius piperatus.* When using 4-methyl catechol as the substrate for PPO, *Km* values were calculated as 0.46 mM in *Lepiota procera* [37], 2.8 mM in *Russula delica* [41], 1.20 mM in *Armillaria mellea*, 9.19 mM in *Lepista nuda*, and 0.51 mM in *Hypholoma fasciculare* [28]. The apparent *Km* values of PPO reported have a large range, which may be due to a variety of reasons: different varieties, different assay methods used, different origins of the same variety, substrate, buffer solution, nutrient sources, purity of the enzyme extract, and different extraction pH.

## 4. Conclusions

In this study, using extraction buffer, ammonium sulfate precipitation, dialysis, and affinity column chromatography, the PPO from the dried *V*. *bombycina* was successfully purified. In addition, the optimal temperature and pH for each of the 4-methyl catechol, pyrogallol, and catechol substrates, as well as *Km* and *Vmax* kinetic values, were provided. To our knowledge, this study provides the first report on the purification and characterization of PPO from dried *V*. *bombycina*. Purification and characterization of PPO studies from new sources are essential for explicating their biochemical behavior and properties. The dried *V*. *bombycina* PPO that was purified and investigated in this work might offer a new source of enzymes for many industrial applications.

## Figures and Tables

**Figure 1 biology-12-00053-f001:**
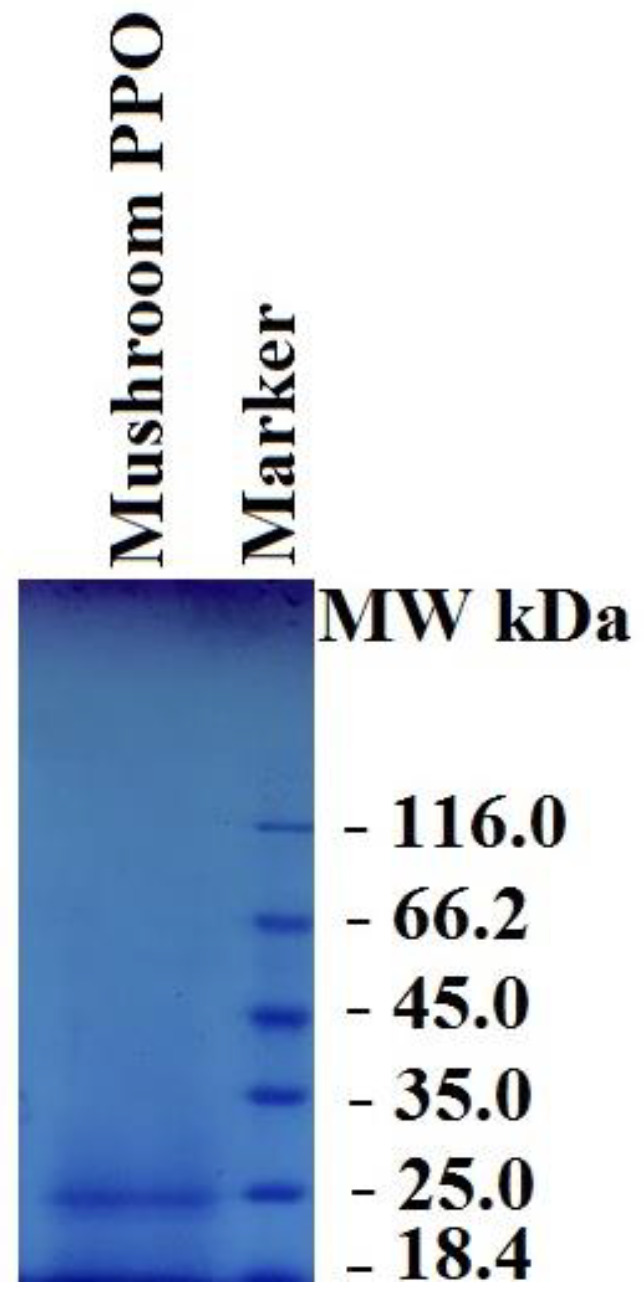
Analysis of the dried *V. bombycina* PPO SDS-PAGE electrophoresis.

**Figure 2 biology-12-00053-f002:**
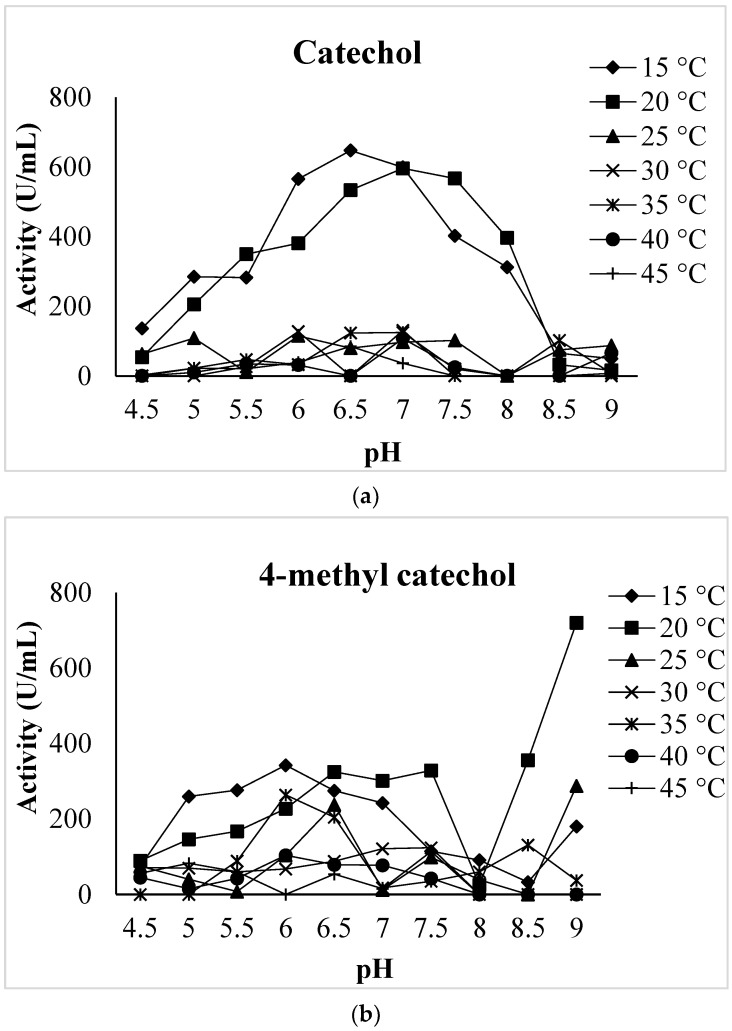
Optimum pH and temperature of dried *V. bombycina* with (**a**) catechol, (**b**) 4-methyl catechol, and (**c**) pyrogallol as substrate.

**Figure 3 biology-12-00053-f003:**
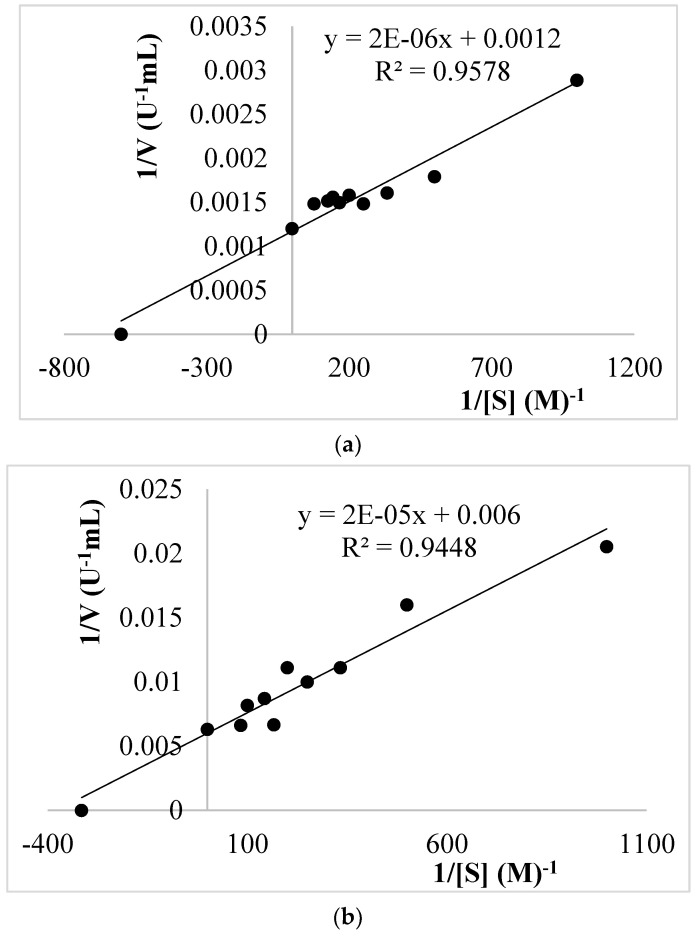
Lineweaver–Burks plots of dried *V. bombycina* PPO with (**a**) catechol, (**b**) 4-methyl catechol, and (**c**) pyrogallol as substrates.

**Table 1 biology-12-00053-t001:** Isolation and purification of PPO from dried *V. bombycina*.

Purification Steps	Volume (mL)	Activity (U/mL)	Total Activity	Total Protein (mg)	Specific Activity (U/mg Protein)	Purification Fold
Crude Extract	36	616.25	22,185.00	13.0138	1704.73	
Ammonium sulfate precipitation	10	488.75	4887.50	9.7475	501.41	0.29
Dialysis	10	1290.00	12,900.00	8.5905	1501.65	0.88
Affinity chromatography	2	275.00	550.00	0.0095	57,698.99	33.85

**Table 2 biology-12-00053-t002:** Optimum pH and temperature values for the dried *V. bombycina* PPO with the substrates.

Substrates	Optimum pH	Optimum Temperature (°C)	Activity (U/mL)
Catechol	6.5	15	647.50
4-methyl catechol	9.0	20	720.00
Pyrogallol	8.0	15	26,692.50

**Table 3 biology-12-00053-t003:** *Km*–*Vmax* values for the dried *V. bombycina* PPO with the substrates.

Substrates	*Km* (mM)	*Vmax* (U/mL)	*Vmax*/*Km*
Catechol	1.67	833.33	500.00
4-methyl catechol	3.17	158.73	50.00
Pyrogallol	2.67	3333.33	1250.00

## Data Availability

The data obtained in this study are available upon request from the corresponding author.

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
