# Peer review of "Purification and Properties of Polyphenol Oxidase of Dried Volvariella bombycina"

_biology, 2022, doi:10.3390/biology12010053_

Round 1

Reviewer 1 Report (Previous Reviewer 2)

The reviewer is satisfied with the improved revision.

Author Response

Thank you so much for your valuable comments and contributions about MS.

Reviewer 2 Report (Previous Reviewer 3)

Please consider the following comments:

1. the statistical section is missed. please add

2. please explain why the activity of Specific activity (U/mg protein) of Crude Extract is more than Ammonium sulfate precipitation.

Author Response

Point 1: The statistical section is missed. please add

Response 1: There are other articles on enzyme studies that have not performed the statistical analysis. In this study, no statistical analysis was performed because the differences between the treatments were too obvious.

Point 2: Please explain why the activity of Specific activity (U/mg protein) of Crude Extract is more than Ammonium sulfate precipitation.

Response 2: Since there are other enzymes catalyzing the same substrate as PPO in the crude extract, it is possible that the ammonium sulfate step rather than the crude extract step may have a higher specific activity.

Reviewer 3 Report (New Reviewer)

Comments and Suggestions for Authors

The results support the principal objective, the characterization properties of PPO from dried V. bombycina. However, However, in the title, should be specified what types of properties are described for the PPO. For example, the kinetic and electrophoretic properties.

The paper needs to be checked carefully for typos and grammatical errors.

L80-82. The phrase “Additionally, PPO is commonly utilized in several fields, including  healthcare and medicine. It is now the focus of attempts to produce new medications to treat Alzheimer's and Parkinson's disease is repeated. It is the same idea in L78-79.

L134. What type of Thermo molecular weight marker? Stained or unstained?

L160-162. The phrase should move to the section Dyalisis.

L178. I suggest using the word concentration instead of level.  L200. The reference for the Effects of pH on dried V. bombycina PPO activity is missing.  L221. I suggest change the phrase for: The molecular weight of PPO was estimated with the protein molecular weight marker (116.0, 66.2, 45.0, 35.0, 25.0, and 18.4 kDa).

The quality of figure 1 should be enhanced. Can you provide the original SDS-PAGE gel electrophoresis of the PPO enzyme? What concentration of purified protein was analyzed on the gel?

L277. Figure legend 2 should be improved. It is not the SDS-PAGE analysis of the molecular weight marker.

Author Response

Point 1: The results support the principal objective, the characterization properties of PPO from dried V. bombycina. However, However, in the title, should be specified what types of properties are described for the PPO. For example, the kinetic and electrophoretic properties.

Response 1: We wanted to keep the title short. It is considered to be more convenient this way.

Point 2: The paper needs to be checked carefully for typos and grammatical errors.

Response 2: In this regard, there will be support from MDPI.

Point 3: L80-82. The phrase “Additionally, PPO is commonly utilized in several fields, including  healthcare and medicine. It is now the focus of attempts to produce new medications to treat Alzheimer's and Parkinson's disease “is repeated. It is the same idea in L78-79.

Response 3: Desired revisions are included in the manuscript.

Point 4: L134. What type of Thermo molecular weight marker? Stained or unstained?

Response 4: Desired revisions are included in the manuscript.

Point 5: L160-162. The phrase should move to the section Dyalisis.

Response 5: Desired revisions are included in the manuscript.

Point 6: L178. I suggest using the word concentration instead of level.  L200. The reference for the Effects of pH on dried V. bombycina PPO activity is missing.  L221. I suggest change the phrase for: The molecular weight of PPO was estimated with the protein molecular weight marker (116.0, 66.2, 45.0, 35.0, 25.0, and 18.4 kDa).

Response 6: Desired revisions are included in the manuscript.

Point 7: The quality of figure 1 should be enhanced. Can you provide the original SDS-PAGE gel electrophoresis of the PPO enzyme? What concentration of purified protein was analyzed on the gel?

Response 7: Desired revisions are included in the manuscript.

Point 8: L277. Figure legend 2 should be improved. It is not the SDS-PAGE analysis of the molecular weight marker.

Response 8: Desired revisions are included in the manuscript.

This manuscript is a resubmission of an earlier submission. The following is a list of the peer review reports and author responses from that submission.

Round 1

Reviewer 1 Report

Dear authors,

I had a great opportunity to review the research manuscript entitled "Purification and Properties of Polyphenol Oxidase of Dried 2
Volvariella bombycina ".

Please add the originalty of this work ( for the first time, we decided to...)

The subject is original but all experiments needs 3 replicates to confirm the data  I recommend authors to improve the discussion and the English also should be too much improved for example ( The aim of this work was to extraction, purification, and characterization of PPO 86 from dried V. bombycina!!). Dialysis!!   The protocol of The activity of enzyme is not clear !! is there positive and negative control? 

Good luck

Author Response

Thank you very much for your valuable comments concerning our manuscript. I have already made the all corrections requested by you on the manuscript carefully.

Reviewer 2 Report

The manuscript presents the purification and characterization of polyphenol oxidase from Volvariella bombycina., including the determination of kinetic parameters, optimum temperature and pH range, and substrate specificity. However, the lack of sequence information encoding the enzyme reduces the significance of the work and makes the judgement of novelty difficult. The exploration of gene information is essential for offering a deeper understanding on enzymology nowadays. Moreover, the manuscript defined the enzyme activity unit as a change in the absorbance of 0.001 mL-1min-1, causing that the data cannot be directly comparable with those of other researchers. The conventional way to define the activity unit is based on the utilization of substrate or the formation of product. Therefore, the quality of the manuscript might not thoroughly meet the requirements of the journal.

Author Response

(The authors gave the same response as above.)

Reviewer 3 Report

Aim of the study was to purify PPO from Volvariella bombycina

1.  please explain application of PPO in medical field

Detailed comments:

What are the main claims of the paper and how significant are they?
PPO was purified and characterized for the first time from a dried wild edible and medicinal mushroom (V. bombycina). This enzyme extracted from fungi is different from plants, but its exact role is not known. As a result, the ability to purify PPO enzyme from the fungus can lead to a better understanding of its function in the future (please see 10.1016/j.phytochem.2006.08.006)

How does the paper stand out from others in its field?
This polyphenol oxidase (PPO) enzyme is used in various fields of medicine, and if extracted with high purity, it can be used in the field of cancer treatment (https://doi.org/10.3390/molecules27051515).

Are the claims novel? If not, which published papers compromise novelty?
According to my review, I did not find an article related to the specific extraction of this enzyme from this mushroom genus.

Are the claims convincing? If not, what further evidence is needed?
The purpose of this research for the first time was to focus on the purification of the enzyme from the mushroom (V. bombycina), therefore it was able to do the claimed

Are there other experiments or work that would strengthen the paper further?
According to the purpose of the study, which was to purify the enzyme, the methods are sufficient. Unless the evaluation of a therapeutic function is done with the MTT test on a cancer cell to assay its anticancer ability.

How much would further work improve it, and how difficult would this be?

It can be a step forward in proving its anti-cancer properties

Would it take a long time?

no-from 2weeks to one month

Are the claims appropriately discussed in the context of previous literature?

yes.

Author Response

(The authors gave the same response as above.)
